# Virulence of Shigatoxigenic and Enteropathogenic *Escherichia coli* O80:H2 in *Galleria mellonella* Larvae: Comparison of the Roles of the pS88 Plasmids and STX2d Phage

**DOI:** 10.3390/vetsci10070420

**Published:** 2023-06-29

**Authors:** Rie Ikeda, Fanny Laforêt, Céline Antoine, Mare Adachi, Keiji Nakamura, Audrey Habets, Cassandra Kler, Klara De Rauw, Tetsuya Hayashi, Jacques G. Mainil, Damien Thiry

**Affiliations:** 1Veterinary Bacteriology, Department of Infectious and Parasitic Diseases, Faculty of Veterinary Medicine, Center for Fundamental and Applied Research for Animals and Health (FARAH), University of Liège, B-4000 Liege, Belgium; rikeda@uliege.be (R.I.); fanny.laforet@uliege.be (F.L.); celine.antoine@uliege.be (C.A.); scc01001@st.osakafu-u.ac.jp (M.A.); a.habets@eurogentec.com (A.H.); cassandra.kler@hotmail.fr (C.K.); jg.mainil@uliege.be (J.G.M.); 2Department of Bacteriology, Faculty of Medical Science, Kyushu University, Fukuoka 812-8582, Japan; nakamura.keiji.046@m.kyushu-u.ac.jp (K.N.); hayashi.tetsuya.235@m.kyushu-u.ac.jp (T.H.); 3Belgium National Reference Center of STEC (NRC STEC), Universitair Ziekenhuis Brussel (UZ Brussel), Vrije Universiteit Brussel (VUB), B-1090 Brussels, Belgium; klara.derauw@azstlucas.be

**Keywords:** *Galleria mellonella* larvae, virulence, Shigatoxigenic *Escherichia coli*, enteropathogenic *Escherichia coli*, O80:H2, O80:non-H2, pS88 plasmid, STX2d phage

## Abstract

**Simple Summary:**

Following the “Replacement/Reduction/Refinement” policy, insects are proposed to replace mammals and birds as experimental models to study the virulence of bacterial pathogens and to identify their virulence properties. The aim of this study was to assess in larvae of the *Galleria mellonella* moth the virulence of the Shigatoxigenic and enteropathogenic *Escherichia coli* O80:H2 and the respective roles of two virulence properties: the pS88 plasmid-encoded invasiveness properties and the phage-encoded Shiga toxin 2d. The objectives were to compare: (i) the virulence of bovine Shigatoxigenic and enteropathogenic *E. coli* O80:H2; (ii) the roles of the pS88 plasmid and Shiga toxin 2d-encoding phage; (iii) the virulence of *E. coli* O80:H2 and O80:non-H2. The results and the conclusions are: (i) *E. coli* O80:H2 and O80:non-H2 are lethal at log5 and log6 concentrations; (ii) the pS88 plasmids are partially responsible for the virulence of *E. coli* O80:H2; (iii) the phage-encoded Stx2d toxin is entirely responsible for the virulence of the Shigatoxigenic *Escherichia coli* O80:H2; (iv) the virulence properties of *E. coli* O80:non-H2 could not be identified. As a general conclusion, *G. mellonella* larvae represent a useful model to study the virulence of bacterial pathogens but are limited in identifying their virulence properties.

**Abstract:**

The invasiveness properties of Shigatoxigenic and enteropathogenic *Escherichia coli* (STEC and EPEC) O80:H2 in humans and calves are encoded by genes located on a pS88-like ColV conjugative plasmid. The main objectives of this study in larvae of the *Galleria mellonella* moth were therefore to compare the virulence of eight bovine STEC and EPEC O80:H2, of two *E. coli* pS88 plasmid transconjugant and STX2d phage transductant K12 DH10B, of four *E. coli* O80:non-H2, and of the laboratory *E. coli* K12 DH10B strains. Thirty larvae per strain were inoculated in the last proleg with 10 μL of tenfold dilutions of each bacterial culture corresponding to 10 to 10^6^ colony-forming units (CFUs). The larvae were kept at 37 °C and their mortality rate was followed daily for four days. The main results were that: (i) not only the STEC and EPEC O80:H2, but also different *E. coli* O80:non-H2 were lethal for the larvae at high concentrations (from 10^4^ to 10^6^ CFU) with some variation according to the strain; (ii) the Stx2d toxin and partially the pS88 plasmid were responsible for the lethality caused by the *E. coli* O80:H2; (iii) the virulence factors of *E. coli* O80:non-H2 were not identified. The general conclusions are that, although the *Galleria mellonella* larvae represent a useful first-line model to study the virulence of bacterial pathogens, they are more limited in identifying their actual virulence properties.

## 1. Introduction

*Escherichia coli* (*E. coli*) is a Gram-negative bacterial species and a member of the human and animal intestinal microbiota. Although most *E. coli* strains are harmless to their hosts, some strains have acquired genes encoding virulence properties and can cause disease in both humans and animals: they are generically called pathogenic *E. coli* [1].

Pathogenic *E. coli* are classified into two main groups: diarrheagenic *E. coli* or DEC and extraintestinal *E. coli* or ExPEC. ExPEC comprise different invasive *E. coli* (septicemic or SePEC, neonatal meningitis-associated or NMEC) and uropathogenic *E. coli* (UPEC) whereas DEC are subdivided into six main pathotypes: enterotoxigenic (ETEC), enteroinvasive (EIEC), enteropathogenic (EPEC), Shigatoxigenic (STEC), enteroaggregative (EAEC), and Diffusely Adherent (DAEC) *E. coli* [1,2,3,4].

Moreover, “hybrid” pathogenic *E. coli* have been described that combine the properties of different pathotypes. One recent dramatic example is the *E. coli* O104:H4 that caused a short-lived outbreak of diarrhea and hemolytic–uremic syndrome in humans in Germany in 2011 and combined the typical properties of EAEC and STEC [5,6]. Nevertheless, the most frequent examples are still today the enterohemorrhagic *E. coli* (EHEC) that emerged in humans in the year 1980 and produced both the Attaching-Effacing (AE) lesion on enterocytes typical of EPEC and one or two Shiga toxins (Stx) typical of STEC [1,3]. Since the EHEC nomenclature is considered obsolete by EFSA [7], they will be named “Attaching-Effacing STEC” (AE-STEC) in this manuscript, as previously proposed [6]. The AE lesion is encoded by genes grouped together on one chromosomal pathogenicity island (Locus of Enterocyte Effacement or LEE) while the Stx are encoded by *stx* phage-located genes [1,3,4]. Two Stx families have been described: Stx1 with three subtypes, (Stx1a, Stx1c, Stx1d) and Stx2 with up to 11 subtypes so far (Stx2a to Stx2k) [8,9,10,11].

The most frequent and pathogenic AE-STEC serotypes in humans are O26:H11, O103:H2, O111:H-, O121:H9, O145:H- and O157:H7, and most frequently the contamination sources are meat, dairy, or vegetable foods contaminated with feces of ruminants, especially cattle, that are healthy carriers in their intestines. Nevertheless, other so-called “minor” serotypes can emerge from time to time and become epidemiologically important in some countries [3,12].

Serotype O80:H2 represents one of these “minor” serotypes that has been emerging since 2010 in France and other Western European countries in humans suffering (bloody) diarrhea, hemolytic–uremic syndrome, and bacteremia [13,14,15,16,17,18]. These AE-STEC O80:H2 combine not only the pathogenic properties of EPEC and STEC, but also of SePEC since they can invade the blood stream and colonize internal organs. The invasiveness properties are encoded by genes located on a pS88-like ColV conjugative plasmid [15,16,18,19].

In addition to their presence in humans, not only AE-STEC, but also EPEC O80:H2 have been emerging since 2009 in young calves in Belgium suffering from diarrhea and only very occasionally from septicemia [20,21]. Genetic analysis confirmed that calf AE-STEC and EPEC O80:H2 are highly related to human AE-STEC O80:H2 by their virulotypes, including the presence of the *eaeξ* intimin-encoding gene on the LEE and of a pS88-like ColV plasmid [21]. Ca. 80% of the bovine AE-STEC harbor the *stx2d* gene like the great majority of human AE-STEC O80:H2, while the remaining bovine and human AE-STEC O80:H2 harbor the *stx1a* or the *stx2a* gene [17,21,22]. Nevertheless, a difference in the virulotypes of AE-STEC and EPEC O80:H2 exists: the majority of *stx2d* AE-STEC harbor the *etsC* and *iucC* genes associated with the pS88 plasmid, but not the *cma* and *iha* chromosomal genes, while the EPEC and the *stx1a* and *stx2a* AE-STEC have the opposite gene profile [21]. Phylogenetically, the different AE-STEC and EPEC O80:H2 form different closely related sub-lineages in the single nucleotide polymorphism (SNP)-based phylogenetic tree and the bovine and human AE-STEC are intermixed in the same sub-lineages. The *stx2d*-positive AE-STEC group together, while the other AE-STEC are more closely related to EPEC [21].

No in vivo study has so far been performed to compare the respective role(s) of the AE lesion, Stx toxin, and pS88-encoded properties in the pathogenicity of AE-STEC and EPEC O80:H2 in humans and calves. Following the 3R policy (Replacement, Reduction and Refinement) [23], insects have been proposed as infectious models to replace mammals and birds for in vivo testing of not only virulence, but also therapy and prophylaxis of bacterial and viral pathogens [24,25]. Of them, larvae of the *Galleria mellonella* moth are frequently used to study bacterial species, including different pathotypes of *E. coli* [25,26,27,28]. Besides their low cost and the possibility of testing multiple groups of larvae in a short time, *G. mellonella* larvae possess an innate immune system similar to mammals and can be maintained at 37 °C, like the bacterial pathogens of mammals [25,29]. So, it was recently observed that the STX2d phage and the type 3 secretion system (T3SS) contribute to the virulence of AE-STEC O80:H2 and EPEC O127:H6, respectively, in *G. mellonella* larvae [25,30,31], while the contribution of the T3SS to the virulence of AE-STEC O157:H7 is less clear [32,33]. However, no study has been performed yet to assess the role of the pS88 plasmids of AE-STEC and EPEC O80:H2.

Therefore, the aim of this study in *G. mellonella* larvae was to compare the virulence of: (i) Belgian calf AE-STEC and EPEC O80:H2 harboring different *stx* genes and pS88 plasmids; (ii) laboratory *E. coli* K12 pS88 plasmid transconjugant and STX2d phage transductant; (iii) Belgian bovine AE-STEC and EPEC O80:H2 and *E. coli* O80:non-H2.

## 2. Materials and Methods

### 2.1. Bacterial Strains

The virulence of 11 wild-type Belgian bovine *E. coli* O80 strains was studied in *Galleria mellonella* larvae (Table 1): eight AE-STEC and EPEC O80:H2 strains with different virulotypes isolated from diarrheic calves at ARSIA (“Association Régionale de Santé et d’Identification animale”) and three *E. coli* O80:H45 and O80:H6 strains isolated from healthy adult cattle [21,34]. The genomes of six AE-STEC and EPEC O80:H2 and of the three *E. coli* O80:H45 and O80:H6 strains were previously sequenced and analyzed. The remaining two AE-STEC O80:H2 were identified by PCR, as previously described [21], and were genome sequenced for confirmation (see Section 2.2). One of the AE-STEC O80:H2 (EH3320/SES3090) had been previously tested in larvae of *G. mellonella* [30] and was the O80:H2 positive control.

The laboratory *E. coli* K12 DH5α strain and one *E. coli* O80:H26 strain of the laboratory serotype collection were the non-O80:H2 negative and positive controls, also based on previously published results in larvae of *G. mellonella* [30]. One *E. coli* O78:H4 strain also of the laboratory serotype collection was added as a control strain after genome sequencing and preliminary testing in *G. mellonella* larvae (see Section 3.1).

Laboratory *E. coli* K12 DH10B and MG1655 strains were used in the pS88 plasmid conjugation and the former was also used in the STX phage transduction. The laboratory *E. coli* K12 strains harbor the genes for the production of the O16 surface antigen although they are phenotypically rough [35].

### 2.2. Genome Sequencing

The genomes of the two not-yet-sequenced AE-STEC O80:H2 and of the *E. coli* O80:H26 and O78:H4 were sequenced and analyzed to confirm the presence or absence of *stx*, *eae*, pS88-located and *cma*/*iha* genes using Virulence Finder 2.0 (https://cge.food.dtu.dk/services/VirulenceFinder/; accessed on 23 March 2023) and their O:H serotypes using SeroType Finder 2.0 (https://cge.food.dtu.dk/services/SerotypeFinder/; accessed on 23 March 2023), as previously described [21,34]. Sequencing data were submitted as NCBI BioProject PRJNA973567. The Genbank accession numbers are SAMN35130762 (O78C), SAMN35130763 (EH3161), SAMN35130764 (SES5320) and SAMN35130765 (SES5363).

### 2.3. Construction of DH10B Transconjugant and Transductant

The pS88 plasmid conjugation strategy was based on the presence of the ColV-encoding *cva* operon on the pS88 plasmid of bovine AE-STEC and EPEC O80:H2 [21,22,23,24,25,26,27,28,29,30,31,32,33,34,35,36]. The recipient strain was the *E. coli* K12 DH10B strain harboring the recombinant plasmid pAuto-ColV-Switch1.0 carrying a synthetic ColV-encoding locus (received from Syngulon Company, Seraing, Belgium; https://syngulon.com). The *cvaC* ColV peptide-encoding gene is, however, disrupted by the insertion of an ampicillin resistance cassette (Amp^R^) while the other genes of the locus remain functional. Therefore, the *E. coli* K12 DH10B strain is unable to produce the ColV peptide but remains immune to ColV since the *cvi* gene is intact.

To choose the donor strain, all 33 Belgian bovine AE-STEC and EPEC O80:H2 previously identified [21] were phenotypically tested for their antibiotic resistance profile by the disk diffusion assay [37] against 12 antibiotics on Mueller–Hinton (AxonLab, Machelen, Belgium) agar plates: ampicillin/amoxicillin (20 μg), amoxicillin/clavulanic acid (20 μg + 10 μg), cefoxitin (30 μg), ceftiofur (30 μg) and cefquinome (30 μg) (β lactams), kanamycin (30 μg) and gentamicin (30 μg) (aminosides), nalidixic acid (30 μg) and enrofloxacin (5 μg) (<fluoro>quinolones), florfenicol (30 μg) (phenicols), doxycycline (30 μg) (tetracyclins), and trimethoprim/sulfamethoxazole (1.25 μg + 23.75 μg) (AxonLab, Machelen, Belgium). According to the EUCAST CASFM and CASFMVET 2020 reference of values (https://www.sfm-microbiologie.org/2020/10/02/casfm-eucast-v1-2-octobre-2020/; https://www.sfm-microbiologie.org/2020/09/09/casfm-veterinaire-2020/; accessed on 15 February 2021), nine strains were sensitive to ampicillin. Of them, the AE-STEC EH2282 strain (Table 1) isolated in 1987 [20] was resistant to only three of the tested antibiotics (kanamycin, doxycycline, trimethoprim/sulfamethoxazole) compared to the other ampicillin-sensitive strains and was therefore chosen as the donor strain.

The pS88 plasmid conjugation was performed on Luria–Bertani (LB; VWR Chemicals, Leuven, Belgium) agar plates [38] by mixing 100 μL of an 8 h growth of the donor strain in 5 mL of LB broth and 500 μL of an 8 h growth of the recipient strain in LB broth with 100 μg/mL ampicillin. After overnight incubation at 37 °C, full loops of the macrocolony were streaked on 10 LB agar plates with 100 μg/mL ampicillin and incubated overnight at 37 °C. Isolated colonies were transferred to two 96-well microtiter plates containing 200 μL LB broth with 100 μg/mL ampicillin and grown overnight at 37 °C. The following day the colonies were transferred onto LB agar plates with 100 μg/mL ampicillin covered with a layer of the ColV-sensitive *E. coli* K12 MG1655 strain harboring the pKK223-3 plasmid [39] carrying an ampicillin resistance-encoding gene (received from Syngulon Company, Seraing, Belgium; https://syngulon.com). The colonies inhibiting the growth of the *E. coli* K12 MG1655 strain were sub-cultured in LB broth with 100 μg/mL ampicillin of which two mL were transferred into two CRYO tubes (Greiner Bio-One, Frickhausen, Germany) with two mL of sterile 80% glycerol that were stored at −20 °C and −80 °C, respectively, until further use.

These colonies were subsequently grown on LB agar plates with 100 μg/mL ampicillin and tested with PCR for the serotype O80-, H2-, and O16-encoding genes, the *cvaC* wild-type and Amp^R^ cassette-inserted genes and the pS88 plasmid-located *hlyF* gene (Table 2), using the FASTGENE2x Optima Hotstart kit (Nippon Genetics, Filter service, Eupen, Belgium) after DNA extraction from one colony by boiling [34].

STX2d phage transductants were constructed using the *E. coli* K12 DH10B strain free of the recombinant plasmid pAuto-ColV-Switch1.0 (received from the Syngulon Company, Seraing, Belgium; https://syngulon.com) as the recipient strain, as previously described [30]. STX2d phages were induced by UV radiation and isolated from the three bovine *stx2d* AE-STEC O80:H2 EH3155, EH3160 and EH3320/SES3090 strains. The transductant candidates were confirmed with a *stx2d* gene quantitative (q) PCR.

### 2.4. In Vivo Assay: The Galleria Mellonella Model

At first, the growth curve of all 17 *E. coli* strains tested in larvae of *G. mellonella* (eight *E. coli* O80:H2, two *E. coli* O80:H45, one *E. coli* O80:H6, the two *E. coli* O80:H26 and O78:H4, the two *E. coli* K12 DH5a and DH10B strains, and the two *E. coli* K12 DH10B pS88 plasmid-conjugated and STX2d phage-transduced strains) was followed by comparing the optical density of an LB broth culture at a wavelength of 600 nm (OD_600_) and the number of colony-forming units (CFU) after plating 10 μL on LB agar and overnight growth at 37 °C. An OD_600_ between 0.2 and 0.35 corresponded to a concentration of 10^8^ CFU/mL depending on the strain. Then, 10 μL of an overnight culture of each strain in LB broth at 37 °C was transferred to fresh LB broth that was incubated at 37 °C until reaching the appropriate OD_600_ corresponding to a concentration of 10^8^ CFU/mL.

Thirty larvae per strain divided into three groups of 10 larvae (Animal Confort, Loncin, Belgium) were inoculated in the last proleg, as previously described [30], with 10 μL of each of the tenfold dilutions of the bacterial culture corresponding to 10 to 10^6^ CFU using an automatic injector (ColeParmer, Vernon Hills, IL, USA). In addition, 30 larvae were injected with 10 μL of PBS as a control group. The larvae were kept at 37 °C and their mortality rate was followed daily for four days. In parallel, back-titration was performed to confirm the actual inoculation dose by streaking 10 μL of each dilution on LB agar plates and incubating them overnight at 37 °C.

### 2.5. Statistical Analysis

All statistical analyses were performed using R software with “Rcmdr v2.6-0” and “survival” packages (https://www.john-fox.ca/RCommander/index.html; accessed on 1 April 2023) after pooling the different groups of larvae per strain, virulotype, or serotype. Kaplan–Meier curves were created to assess the survival of the larvae according to the inoculation doses of each bacterial strain. Log-rank tests were carried out to highlight significant differences in survival rates between the groups of larvae inoculated with the different concentrations (log1 to log6) of each *E. coli* strain and the PBS group. Hazard ratios (HRs) with a 95% confidence interval (HR-95%) were calculated to give the relative measure of the risk factor for death for the larvae when comparing the same concentration of two different *E. coli* strains or groups of strains, the reference and the test strains: when the HR-95% values included the value 1, the risk factor for death was the same with both strains; when both HR-95% values were >1 or <1, the risk factor for death was higher or lower, respectively, with the test strain(s) compared to the reference strain(s). The significant thresholds of 0.05, 0.01, and 0.001 were applied for all statistical analysis.

## 3. Results

### 3.1. Virulotypes and Serotypes of the Four Genome Sequenced E. coli Strains

The genome sequencing and analysis confirmed the virulotypes and serotypes of the two AE-STEC O80:H2 SES5320 and SES5363 strains using Virulence Finder 2.0 and SeroType Finder 2.0 (Table 1 and Table 3). The *eaeξ* and *stx1a* genes and several genes located on the pS88 plasmid, including the specific *hlyF* gene, were detected. The pS88 plasmid-located *etsC*/*iucC* genes were detected in the AE-STEC O80:H2 SES5363 strain, but not in the AE-STEC O80:H2 SES5320 strain. Conversely, the chromosomal *cma*/*iha* genes were detected in the AE-STEC O80:H2 SES5320 strain, but not in the AE-STEC O80:H2 SES5363 strain. Therefore, three *stx1a* AE-STEC, three *stx2d* AE-STEC, and two EPEC O80:H2 strains with different pS88 plasmid and chromosomal gene profiles (Table 1 and Table 3) were tested in *G. mellonella* larvae.

The serotypes of the *E. coli* O80:H26 and O78:H4 of the laboratory serotype collection were also confirmed (Table 1 and Table 3). The *stx* genes and the LEE-located genes were not detected, identifying them as neither AE-STEC nor EPEC. Conversely, the pS88 plasmid-located genes, including the *etsC*/*iucC* genes, were detected in the *E. coli* O78:H4 strain, but not in the *E. coli* O80:H26 strain (Table 1 and Table 3), whereas the *cma*/*iha* chromosomal genes were not detected. Since the *E. coli* O78:H4 was pathogenic for larvae of *G. mellonella* in preliminary testing and harbored one pS88-like plasmid, this strain was added as a non-O80 pS88 plasmid-positive control.

### 3.2. Identification of Transconjugant and Transductant

After growth in LB broth with 100 μg/mL ampicillin in microtiter plates, 50 colonies were transferred onto LB agar plates with 100 μg/mL ampicillin covered with a layer of the ColV-sensitive ampicillin-resistant *E. coli* K12 MG1655 strain. Eleven of these 50 colonies inhibited the growth of the *E. coli* K12 MG1655 strain and gave two amplification fragments (314 bp and 1555bp; Table 2) with the PCR for the *cvaC* wild-type and Amp^R^ cassette-inserted genes, respectively. Of the 11 transconjugant candidates, two also tested positive with the PCR for the O16-encoding genes and the pS88 plasmid-located *hlyF* gene and negative with the PCR for the serotype O80- and H2-encoding genes. These two transconjugant candidates (F5 and D4) were sub-cultured on two LB agar plates with 100 μg/mL ampicillin. Ten colonies from each agar plate were chosen and re-tested with the same PCR. Of these 20 colonies, 12 once more gave unambiguous PCR results and 1 colony from the F5 transconjugant candidate was chosen for testing in *G. mellonella* larvae.

The STX2d phages isolated from the AE-STEC O80:H2 EH3155 and EH3160 strains, but not from the EH3320/SES3090 strain, produced plaque lysis on the *E. coli* K12 DH10B strain. Either phage and the *E. coli* K12 DH10B strain were mixed in LB broth and incubated overnight at 37 °C. The transductant candidates were recovered by centrifugation, re-suspended in LB broth, and spread on STEC agar plates. After overnight incubation at 37 °C, one colony from each plate was randomly picked up and confirmed by qPCR targeting the *stx2d* gene. These colonies were sub-cultured three times on STEC agar to confirm the stability of the transduced phages. At each stage, the transductant candidates were confirmed by the qPCR targeting the *stx2d* gene. *E. coli* K12 DH10B transductant from AE-STEC O80:H2 EH3160 strain was chosen for studies in *G. mellonella* larvae.

### 3.3. Virulence of E. coli Strains in G. mellonella Larvae

#### 3.3.1. *E. coli* Non-O80:H2 Control Strains

Of these three control strains (*E. coli* K12 DH5α, *E. coli* O80:H26, and *E. coli* O78:H4), the *E. coli* O78:H4 strain was the most highly virulent, killing almost all larvae within 24 h post-inoculation (HPI) even at log1 concentration (*p*-value < 0.001), while more than 80% of larvae inoculated with the *E. coli* K12 DH5α strain still survived at 96 HPI at log6 concentration (Figure 1; Appendix A). The *E. coli* O80:H26 strain was also virulent, killing about half and 75% of the larvae at log4 and log5 concentrations, respectively, at 96 HPI and all larvae at log6 concentration at 72 HPI (*p*-value < 0.001; Figure 1; Appendix A). The *E. coli* K12 DH5α and O80:H26 strains gave similar results to those previously observed [30].

#### 3.3.2. Comparison of the *E. coli* O80:H2 Strains Belonging to Different Virulotypes

According to the log-rank analysis of Kaplan–Meier curves (Figure 2, Figure 3 and Figure 4; Appendix A), the log5 and log6 concentrations of all eight *E. coli* O80:H2 strains gave significantly different results from the PBS group results with less than 40% survival at 96 HPI (*p*-value < 0.001). The *stx2d* AE-STEC EH3320/SES3090 strain gave similar results to those previously obtained [30]. Although all three *stx2d* AE-STEC were also significantly more lethal compared to the PBS group at the log4 concentration (*p*-value < 0.001) with ca. 40% death at 96 HPI (Figure 3; Appendix A), this was not the case for the *stx1a* AE-STEC and at least for one EPEC (Figure 2 and Figure 4; Appendix A). Conversely, the results with the lowest concentrations (log1 to log3) were much more heterogeneous, even within the same virulotype (Figure 2, Figure 3 and Figure 4; Appendix A), and were not further analyzed. Therefore, the hazard ratios (HRs) and the confidence intervals 95% (HR-95%) were statistically analyzed only for the log5 and log6 concentrations.

At the highest bacterial concentration (log6), almost all larvae were dead at 96 HPI regardless of the virulotype (Figure 2, Figure 3 and Figure 4). However, the *stx2d* AE-STEC strains significantly killed more larvae and more quickly than the EPEC (Figure 3 and Figure 4) with all larvae dead at 72 HPI while a few larvae were still alive at 96 HPI with the EPEC. The HR-95% of the *stx2d* AE-STEC vs. EPEC strains was statistically significant: between 0.44 and 0.88 (*p*-value < 0.01) (Appendix A). Conversely, the results with *stx1a* vs. *stx2d* AE-STEC and with *stx1a* AE-STEC vs. EPEC were not significantly different (Figure 2 and Figure 3; Appendix A).

At the log5 concentration, the *stx2d* AE-STEC strains significantly killed more larvae and more quickly than the EPEC and the *stx1a* AE-STEC, with a survival rate lower than 20% at 96 HPI vs. 10–40% for the different *stx1a* AE-STEC and EPEC strains (Figure 2, Figure 3 and Figure 4). The HC-95% of the *stx1a* AE-STEC vs. *stx2d* AE-STEC was higher than 1 (between 1.17 and 2.19 with a *p*-value < 0.01), while the HC-95% of the *stx2d* AE-STEC vs. EPEC was lower than 1 (between 0.40 and 0.83 with a *p*-value < 0.01) (Appendix A). Conversely, the results of the *stx1a* AE-STEC vs. EPEC were not significant, similar to the log6 concentration.

Since some results could be due to intra-group variations as consequence of different genetic backgrounds, the results of the three *stx2d* AE-STEC, the three *stx1a* AE-STEC, and the two EPEC were intra-virulotype compared (Appendix A). Only the results of the two EPEC strains were statistically different at the log6 concentration. The EPEC EH3322/SES3122 strain harboring a pS88 plasmid carrying the *etsC* and *iucC* genes (pS88++) killed more larvae and more rapidly than the EH3308/SES2973 strain whose pS88 plasmid does not carry these two genes (pS88--) (Figure 4). The HR-95% of the EPEC EH3308/SES2973 pS88-- strain vs. the EPEC EH3322/SES3122 pS88++ strain was statistically significant: between 1.09 and 3.23 (*p*-value < 0.05) (Appendix A).

#### 3.3.3. Comparison of the Role of the pS88 Plasmid and of the STX2d Phage

According to the results described above, the *etsC*/*iucC*-positive pS88 plasmid may play a role in the pathogenicity of some *E. coli* O80:H2 strains in *G. mellonella* larvae. The virulence of one *etsC*/*iucC*-positive pS88 plasmid-conjugated *E. coli* K12 DH10B strain was therefore compared with the virulence of one STX2d phage-transduced *E. coli* K12 DH10B, of the *E. coli* K12 DH10B recipient, and of the AE-STEC O80:H2 EH2282 and EH3160 plasmid and phage donor strains (Table 1).

The log-rank analysis of the Kaplan–Meier curves confirmed that the acquisition by conjugation of the *etsC*/*iucC*-positive pS88 plasmid significantly increased the lethality of the *E. coli* K12 DH10B recipient strain (ca. 25% at 96 HPI), both at log5 (*p*-value < 0.05) and log6 (*p*-value < 0.01) concentrations (Figure 5; Appendix A). However, the HR-95% of the transconjugant vs. the *E. coli* K12 DH10B recipient strain was not statistically significant, neither at log6 nor at log5 (Appendix A). Conversely, the HR-95% of the transconjugant vs. the *stx1a* AE-STEC EH2282 donor strain was highly significant (between 5.74 and 29.37) at log6 (*p*-value < 0.001) and between 4.32 and 23.82 at log5 (*p*-value < 0.001) concentrations (Appendix A). Nevertheless, the HR-95% of the *E. coli* K12 DH10B recipient strain vs. the *stx1a* AE-STEC EH2282 donor strain was even more highly significant: between 12.78 and 153.50 at log6 (*p*-value < 0.001) and between 10.57 and 586.30 at log5 (*p*-value < 0.001) concentrations (Appendix A).

The log-rank analysis of the Kaplan–Meier curves also confirmed that the acquisition by transduction of the STX2d phage significantly increased the lethality of the *E. coli* K12 DH10B recipient strain, at log4 (*p*-value < 0.01), log 5 (*p*-value < 0.001), and log 6 (*p*-value < 0.001) concentrations. The lethality of the transduced *E. coli* K12 DH10B strain was very similar to the lethality of the *stx2d* AE-STEC EH3160 donor strain (Figure 6; Appendix A): less than 20% of the larvae inoculated with the log5 concentration survived at 96 HPI with both transduced and donor strains while all larvae were dead at 72 HPI and at 48 HPI, respectively, at the log6 concentration (Figure 6). Moreover, the HR-95% of the STX2d-transduced *E. coli* K12 DH10B and of the *stx2d* AE-STEC EH3160 donor strains vs. the *E. coli* K12 DH10B recipient strain were similar and also highly significant with HR-95% values between ca. 8 and more than 400 at log5 and between ca. 17 and more than 350 at log6 (Appendix A). Conversely, the HR-95% of the STX2d-transduced *E. coli* K12 DH10B vs. the *stx2d* AE-STEC EH3160 donor strains was not statistically significant (Appendix A).

#### 3.3.4. Comparison with Other *E. coli* O80 Serotypes

In addition to AE-STEC and EPEC O80:H2, the virulence in *G. mellonella* larvae of three pS88 plasmid-negative, non-AE-STEC, non-EPEC *E. coli* O80:H6 and O80:H45 strains isolated from healthy cattle (Table 1 and Table 3) was assessed for comparison with all *E. coli* O80:H2 strains. The control *E. coli* O80:H26 strain was also included in this comparison.

According to the log-rank analysis of the Kaplan–Meier curves (Figure 7; Appendix A), the log5 (*p*-value < 0.001) and log6 (*p*-value < 0.001) concentrations of all three *E. coli* O80:H6 and O80:H45 strains gave significantly different results from the PBS group results with less than 30% and 10% survival at 96 HPI, respectively, while the results with the lower concentrations (log1 to log4) were not consistently statistically significant with more than 70% survival at 96 HPI (Figure 7; Appendix A). The results of the *E. coli* O80:H26 strain were similar to the log5 and log6 concentrations, but the log4 concentration was also statistically significant (*p*-value < 0.001) (Figure 7; Appendix A). Moreover, the HR-95% ratios between these three O80 serotypes were not statistically significant with the exception of the HR-95% of the *E. coli* O80:H6 strain vs. the *E. coli* O80:H26 strain that was between 1.11 and 3.12 (*p*-value < 0.05) at the log6 concentration (Appendix A).

As for the comparison between the results of *E. coli* O80:H6, O80:H26, and O80:H45 strains on the one hand and the *stx1a* AE_STEC, *stx2d* AE-STEC, and EPEC O80:H2 on the other hand, some statistically significant differences were observed, especially between the *E. coli* O80:H6 strain and the AE-STEC O80:H2 strains (Appendix A). For instance, the HR-95% ratios of the *E. coli* O80:H6 strain vs. the *stx1a* and *stx2d* AE-STEC O80:H2 strains were statistically significant at log6: between 0.41 and 0.98 (*p*-value < 0.05) and between 0.31 and 0.76 (*p*-value < 0.01), respectively. At log5, the HR-95% of the *E. coli* O80:H6 strain vs. the *stx2d* AE-STEC O80:H2 strains was also statistically significant: between 0.38 and 0.93 (*p*-value < 0.05). In addition, the HR-95% of the *E. coli* O80:H45 strain vs. the *stx2d* AE-STEC O80:H2 strains was statistically significant at log6 (between 0.49 and 0.97; *p*-value < 0.05), but not at log5 (Appendix A). Conversely, the HR-95% ratios of any *E. coli* O80:non-H2 strain vs. the EPEC O80:H2 strains were not statistically significant (Appendix A).

## 4. Discussion

The emerging AE-STEC O80:H2 are recognized as a multiple hybrid pathotype responsible for enteritis, hemolytic–uremic syndrome, and bacteremia or septicemia in humans and/or young calves [13,15,21]. They indeed harbor not only phage-located *stx* genes with a large majority of *stx2d* gene and/or LEE pathogenicity island-located genes, but also pS88-like ColV plasmids carrying invasiveness-encoding genes [15,21,22]. However, the respective role(s) of the Stx toxins, AE lesion, and pS88-encoded properties in their pathogenicity have not been confirmed by in vivo studies yet. Following the 3R policy (Replacement, Reduction and Refinement) [23], insects have been recommended and have been used for some years as a first step for in vivo testing of pathogenic bacterial species [24,25]. Thus, the main purpose of this study was to assess larvae of the *G. mellonella* moth [24,25,27] as a model to study and compare the virulence of different AE-STEC and EPEC O80:H2 isolated from calves in Belgium [21], and the respective roles of the pS88-like ColV plasmid and STX2d phage.

According to the statistical analysis results, all eight calf *stx1a* AE-STEC, *stx2d* AE-STEC, and EPEC O80:H2 were pathogenic for *G. mellonella* larvae at log5 and log6 concentrations compared to the PBS group (Figure 2, Figure 3 and Figure 4; Appendix A). They were also more pathogenic than the *E. coli* K12 DH5α and DH10B strains, but less than the *E. coli* O78:H4 strain (Figure 1, Figure 5 and Figure 6; Appendix A). Moreover, all three *stx2d* AE-STEC were already lethal for larvae at log4 and more rapidly and intensively at log5 and log6 than the *stx1a* AE-STEC and EPEC (Figure 2, Figure 3 and Figure 4; Appendix A).

The reason for the *stx2d* AE-STEC being more virulent to *G. mellonella* larvae may be related to the production of the Stx2d toxin. Indeed, the lethality rate of larvae with STX2d phage-transduced *E. coli* K12 DH10B strain was significantly higher compared to the *E. coli* K12 DH10B recipient strain and was similar to the AE-STEC O80:H2 EH3160 phage donor strain (Figure 6; Appendix A), as previously reported with the STX2d phage-transduced *E. coli* K12 DH5α strain [30]. The HR-95% ratios of the STX2d phage-transduced *E. coli* K12 DH10B strain and of the EH3160 phage donor strains were also similar and significantly higher when compared with the *E. coli* K12 DH10B recipient strain. These results confirmed that the Stx2d toxin plays an important role in the pathogenicity of *stx2d* AE-STEC O80:H2 in *G. mellonella* larvae, as previously suggested [30].

Although the HR-95% between *stx1a* AE-STEC and *stx2d* AE-STEC was statistically significant at log5, a role for the Stx1a toxin in the lethality of larvae cannot be totally excluded, since the difference was not statistically significant at log6 (Appendix A). The reasons for these conflicting results may be several: (i) low numbers of CFUs of one *stx1a* AE-STEC actually inoculated, especially at log5 (3 × 10^4^ CFU instead of 10^5^ CFU after back-titration; data not shown); (ii) slower multiplication of *stx1a* AE-STEC compared to *stx2d* AE-STEC as a consequence of different general genetic backgrounds of the strains; (iii) delay in the production of the Stx1a toxin compared to the Stx2d toxin; (iv) less efficacious action of the Stx1a toxin in *G. mellonella* larvae compared to the Stx2d toxin. Whatever the actual reason, counting the number of CFUs of *stx1a* AE-STEC and *stx2d* AE-STEC in dead larvae, testing STX1a phage transductant, and/or following the expression of the *stx1a* and *stx2d* genes may answer these questions.

Two other specific properties of the AE-STEC and EPEC O80:H2 are the production of the AE lesion and the invasiveness properties encoded by pS88-like plasmid-located genes [15,18,19,21]. The role of the T3SS responsible for the development of the AE lesion was not assessed in his study but is indeed partially responsible for the lethality of the human EPEC E2348/69 (serotype O127:H6) in *G. mellonella* larvae compared to a mutant in one of the encoding genes, as observed by others [31]. However, the situation seems different for AE-STEC O157:H7 [32,33], possibly because the Stx toxins play a more important role ([30], this study).

As far as the pS88 plasmids are concerned, a role of the *etsC*/*iucC*-positive pS88 plasmid can be hypothesized, according to the statistical analysis of the pathogenicity in *G. mellonella* larvae of the two EPEC O80:H2 (Figure 4; Appendix A). Indeed, the EPEC EH3322/SES3122 strain harboring one *etsC*/*iucC*-positive pS88 plasmid (Table 1 and Table 3) was almost twice as lethal as the EPEC EH3308/SES2973 strain harboring a pS88 plasmid not carrying these two genes at log6 (Appendix A). However, the *etsC*/*iucC*-positive pS88 plasmid transconjugant was not statistically more lethal than the *E. coli* K12 DH10B recipient strain, although the HR-95% lower values were borderline at both log6 (between 0.92 and 12.60) and log5 (between 0.95 and 63.07) concentrations (Appendix A). Nevertheless, this transconjugant was far from being as lethal as the *stx1a* AE-STEC O80:H2 plasmid donor strain (Figure 5; Appendix A). The *etsC* gene is a member of one operon coding for an ABC transporter system while the *iucC* gene is a member of an operon encoding the aerobactin siderophore [43]. Although both *ets* and *iuc* genes are markers of ExPEC, especially of Avian Pathogenic *E. coli* (APEC) and of NMEC, the contribution of the *ets* genes in the pathogenicity of *E. coli* in larvae of *G. mellonella* is still unknown, while contradictory results have been obtained for the *iuc* genes [19,44]. Comparison with the results obtained with one *etsC*/*iucC*-negative pS88 plasmid transconjugant would represent a first step in the understanding of their role.

The role of the pS88 plasmids could actually depend more on the general genetic background of the *E. coli* tested than on the pS88 plasmid virulotype. For instance, the difference observed between the two EPEC O80:H2 strains was not observed between the three *stx2d* AE-STEC or between the three *stx1a* AE-STEC (Appendix A), possibly because the Stx toxins were also here more important virulence factors in *G. mellonella* larvae than the properties encoded by the pS88 plasmid. Moreover, the great majority of AE-STEC and EPEC O80:H2 harboring one *etsC*/*iucC*-positive pS88 plasmid group together in a single nucleotide polymorphism-based phylogenetic tree were negative for the *iha* and *cma* chromosomal genes (Table 3), and vice versa [21]. Nevertheless, no role for the *cma* and *iha* genes in the pathogenicity of *E. coli* O80:H2 in this intrahemocoelic inoculation model of larvae of *G. mellonella* can be proposed at this stage. Indeed, the *cma* gene codes for a colicin degrading the glycan chain of the murein precursor of other *E. coli* cells [45] while the *iha* gene encodes an outer membrane protein, conferring adherence to epithelial cells in culture [46]. Moreover, the *iha* gene is located on the pathogenicity island SPLE1, and its absence can also mean the absence of other genes located on SPLE1 [46,47].

Another striking observation was the human EPEC O127:H6 that was already lethal for *G. mellonella* larvae at a concentration of 5 × 10^3^ CFU, like the *stx2d* AE-STEC O80:H2, while not harboring any *stx* genes or pS88 plasmid [31]. Therefore, testing different pS88 plasmid-cured *E. coli* O80:H2 strains would also help to determine its actual role in *G. mellonella* larvae, as already published in a mammalian model with the S88 NMEC strain. The S88 strain cured of the pS88 plasmid loses almost all virulence in a neonatal rat model of infection while reintroduction of the plasmid restores full virulence [19]. The role of any bacterial property can indeed also depend on the animal model. Testing the S88 strain and its plasmid-cured derivative would also help to confirm *G. mellonella* larvae as an in vivo model to study the virulence of different *E. coli* pathotypes.

Still another example of the importance of the genetic background of the strains was the *E. coli* O78:H4 control strain that harbors an *etsC*/*iucC*-positive pS88 plasmid (Table 1 and Table 3). This *E. coli* O78:H4 strain, however, had the highest lethality and pathogenicity to *G. mellonella* larvae in spite of being neither AE-STEC nor EPEC (Figure 1; Appendix A). Septicemia-associated *E. coli* serotype O78 are frequently isolated from mammals and poultry and one of the important virulence factors is the production of the O78 lipopolysaccharide that displayed anti-complement properties in chickens [4,48]. Since *G. mellonella* larvae have an innate immune system similar to mammals and birds, including the presence of complement-like proteins [25,26,27,28,29], any *E. coli* property conferring enhanced resistance to complement may also increase its virulence in *G. mellonella* larvae.

Another reason could be the availability of and the access to iron in *G. mellonella* larvae and the challenge model. To the authors’ knowledge, there are no data published about the former in *G. mellonella*. About the latter, APEC and NMEC produce several iron-chelation systems, including two encoded by pS88-located genes, the salmochelin (*iroBCDEN* genes) and the aerobactin (*iuc* and *iutA* genes). Although they both can contribute to the virulence of APEC in poultry [44,49], neither is important for the pathogenicity of NMEC strain S88 in the neonatal rat model [19]. Nevertheless, the situation may be different in *G. mellonella* larvae, since the EPEC O80:H2 strain harboring an *iucC*-positive pS88 plasmid was statistically more lethal for *G. mellonella* larvae than the EPEC strain harboring an *iucC*-negative pS88 plasmid (Figure 4; Appendix A).

The role of other genes located on plasmids or on the chromosome, including on pathogenicity islands and/or phages, in the virulence of different *E. coli* strains is beyond any doubt [1,50] but was here illustrated by the results of the four pS88-negative *E. coli* O80:H6, O80:H26, and O80:H45 strains in *G. mellonella* larvae. These four *E. coli* O80:non-H2 strains were indeed statistically significantly lethal for larvae at log5 and log6 compared to the PBS group, like all the *E. coli* O80:H2 strains (Figure 2, Figure 3, Figure 4 and Figure 7; Appendix A). Nevertheless, the *E. coli* O80:H26 strain was already statistically significantly lethal for the larvae at the log4 concentration, as previously reported [30], and like the *stx2d* AE-STEC O80:H2, killed half of the larvae at 96 HPI vs. less than 25% for the other three O80:non-H2 strains (Figure 7; Appendix A).

The *E. coli* O80:H26 actually appeared to be the most lethal and the *E. coli* O80:H6 the least lethal of the *E. coli* O80:non-H2 serotypes for *G. mellonella* larvae. The *E. coli* O80:H26 was, for instance, statistically more lethal than the *E. coli* O80:H6 to *G. mellonella* larvae with the HR-95% of the *E. coli* O80:H6 vs. the *E. coli* O80:H26 statistically significantly > 1 (Appendix A). Moreover, the HR-95% ratios of all AE-STEC and EPEC O80:H2 vs. the *E. coli* O80:H26 strains were not statistically significant (Appendix A). Conversely, the HR-95% of the *stx2d* and *stx1a* AE-STEC O80:H2 vs. the *E. coli* O80:H6 strain were significantly < 1 at the log5 and/or log6 concentrations (Appendix A), while the HR-95% of the *stx2d* but not of the *stx1a* AE-STEC O80:H2 vs. the *E. coli* O80:H45 strains was significantly < 1 but only at the log6 concentration (Appendix A).

All these results confirmed the necessary role of the general genetic background in the virulence of *E. coli* strains to *G. mellonella* larvae. Unfortunately, at this stage, no correlation between the general virulotypes of these four *E. coli* O80:non-H2 strains could be made, although some putative virulence genes related to ExEPEC strains (Table 3) were detected. It is however striking that none of these genes could be detected in the *E. coli* O80:H26 strain (Table 3) that was the most virulent of the *E. coli* O80:non-H2 strains. This represented either our shortage of knowledge on the actual role of several genes of *E. coli* in virulence in *G. mellonella* larvae, or the current limits of the Virulence Finder tool.

## 5. Conclusions

The general conclusions of this study in *G. mellonella* larvae are that: (i) not only the AE-STEC and EPEC O80:H2 but also different *E. coli* O80:non-H2 strains are lethal at high concentrations (log5 and log6 CFU); (ii) the pS88 plasmids, especially the *etsC*/*iucC*-positive pS88 plasmids, are partly responsible for the lethality of the EPEC O80:H2; (iii) the Stx2d toxins are entirely responsible for the lethality of the *stx2d* AE-STEC O80:H2; and (iv) the identity of the virulence factor(s) responsible for the lethality of the *E. coli* O80:non-H2 strains is unknown at this stage.

Identification of these different virulence factors and understanding their respective role(s) are beyond the purpose of this study but could be the goals of future studies with different mutants engineered by, for instance, plasmid curing and allelic exchanges and/or with an in vivo imaging system using bioluminescence or fluorescence microscopy strains [19,31,32,33,51,52,53]. In parallel, comparative in vivo studies with mammalian and avian models are needed to further assess insects, especially larvae of the *G. mellonella* moth, as an in vivo challenge model to study and elucidate bacterial virulence.

## Figures and Tables

**Figure 1 vetsci-10-00420-f001:**
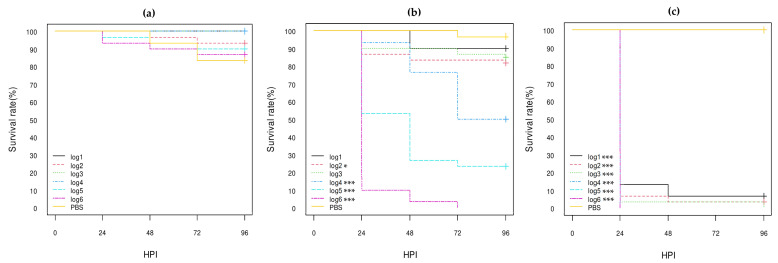
Kaplan–Meier survival curves of control *E. coli* strains: (**a**) K12-DH5α; (**b**) serotype O80:H26 (EH3161 strain); (**c**) serotype O78:H4 (pS88++). HPI = Hour Post-Inoculation. * *p*-value statistically significant at the threshold 0.05. *** *p*-value statistically significant at the threshold 0.001. pS88++: pS88 plasmid carrying the *etsC*/*iucC* genes.

**Figure 2 vetsci-10-00420-f002:**
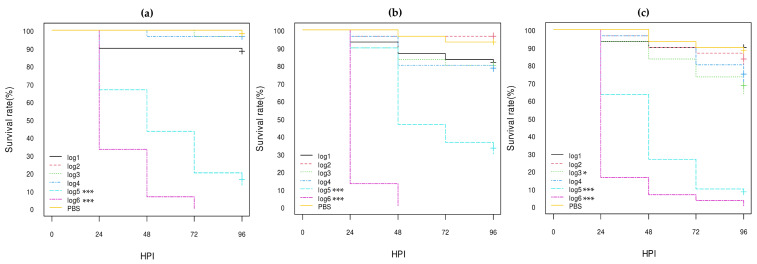
Kaplan–Meier survival curves of *stx1a* AE-STEC O80:H2 strains: (**a**) SES5320 strain (pS88--); (**b**) SES5363 strain (pS88++); (**c**) EH2282 strain (pS88++). HPI = Hour Post-Inoculation. * *p*-value statistically significant at the threshold 0.05. *** *p*-value statistically significant at the threshold 0.001. pS88++: pS88 plasmid carrying the *etsC*/*iucC* genes. pS88--: pS88 plasmid not carrying the *etsC*/*iucC* genes.

**Figure 3 vetsci-10-00420-f003:**
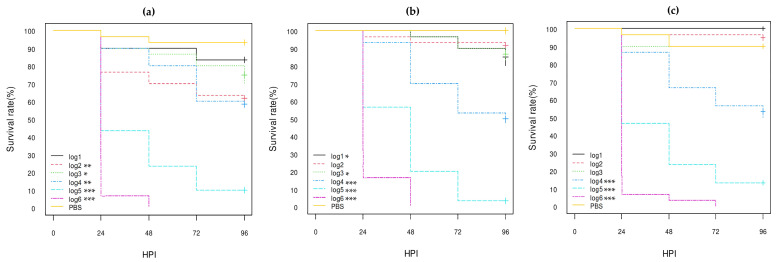
Kaplan–Meier survival curves of *stx2d* AE-STEC O80:H2 strains: (**a**) EH3160 strain (pS88++); (**b**) EH3307/SES2959 strain (pS88--); (**c**) EH3320/SES3090 strain (pS88++). HPI = Hour Post-Inoculation. * *p*-value statistically significant at the threshold 0.05. ** *p*-value statistically significant at the threshold 0.01. *** *p*-value statistically significant at the threshold 0.001. pS88++: pS88 plasmid carrying the *etsC*/*iucC* genes. pS88--: pS88 plasmid not carrying the *etsC*/*iucC* genes.

**Figure 4 vetsci-10-00420-f004:**
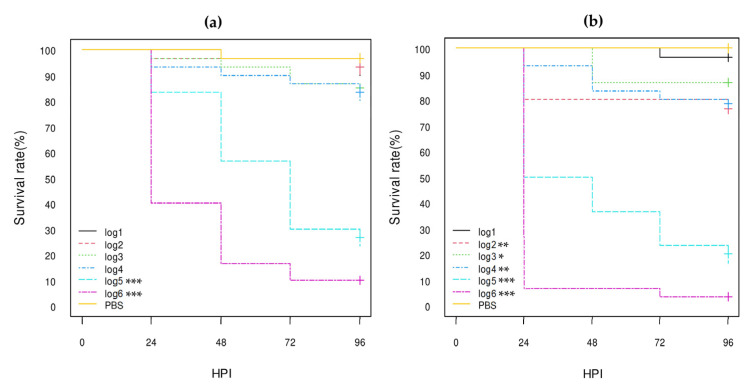
Kaplan–Meier survival curves of EPEC O80:H2 strains: (**a**) EH3308/SES2973 strain (pS88--); (**b**) EH3322/SES3122 strain (pS88++). HPI = Hour Post-Inoculation. * *p*-value statistically significant at the threshold 0.05. ** *p*-value statistically significant at the threshold 0.01. *** *p*-value statistically significant at the threshold 0.001. pS88++: pS88 plasmid carrying the *etsC*/*iucC* genes. pS88--: pS88 plasmid not carrying the *etsC*/*iucC* genes.

**Figure 5 vetsci-10-00420-f005:**
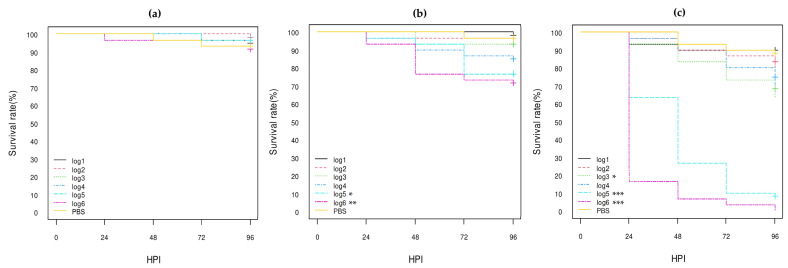
Kaplan–Meier survival curves of: (**a**) K12-DH10B recipient strain; (**b**) K12-DH10B pS88 plasmid-conjugated strain (pS88++); (**c**) *stx1a* AE-STEC O80:H2 EH2282 pS88 plasmid donor strain (pS88++). HPI = Hour Post-Inoculation. * *p*-value statistically significant at the threshold 0.05. ** *p*-value statistically significant at the threshold 0.01. *** *p*-value statistically significant at the threshold 0.001. pS88++: pS88 plasmid carrying the *etsC*/*iucC* genes.

**Figure 6 vetsci-10-00420-f006:**
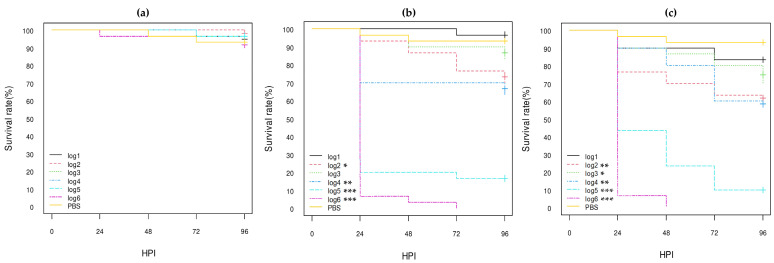
Kaplan–Meier survival curves of: (**a**) K12 DH10B recipient strain; (**b**) K12 DH10B STX2d phage-transduced strain; (**c**) *stx2d* AE-STEC O80:H2 EH3160 STX2d phage donor strain. HPI = Hour Post-Inoculation. * *p*-value statistically significant at the threshold 0.05. ** *p*-value statistically significant at the threshold 0.01. *** *p*-value statistically significant at the threshold 0.001.

**Figure 7 vetsci-10-00420-f007:**
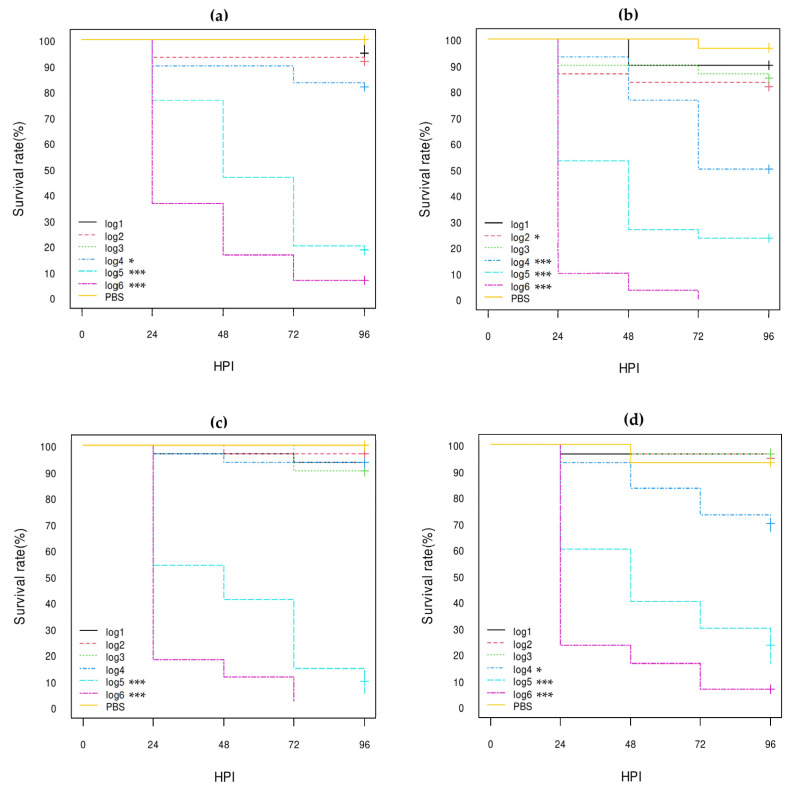
Kaplan–Meier survival curves of *E. coli* O80:non-H2 strains: (**a**) serotype O80:H6 SES6039 strain; (**b**) serotype O80:H26 (EH3161 strain); (**c**) serotype O80:H45 SES5725 strain; (**d**) serotype O80:H45 SES6156 strain. HPI = Hour Post-Inoculation. * *p*-value statistically significant at the threshold 0.05. *** *p*-value statistically significant at the threshold 0.001.

**Table 1 vetsci-10-00420-t001:** Serotypes and virulotypes of the wild-type *E. coli* strains used in this study.

Isolate References ^1^	Serotype	*stx* Genes	*eae* Gene	pS88 Plasmid Virulotype (*iucC*/*etsC* Genes)	Reference of Genome Sequencing
SES5320	O80:H2	1a	ξ	nd/nd ^2^	This study
SES5363		1a	ξ	d/d	
EH2282 ^3,4^		1a	ξ	d/d	[21]
EH3160 ^4^		2d	ξ	d/d	
EH3307/SES2959		2d	ξ	nd/nd	
EH3320/SES3090 ^5^		2d	ξ	d/d	
EH3308/SES2973		nd	ξ	nd/nd	
EH3322/SES3122		nd	ξ	d/d	
SES6039	O80:H6	nd	nd	--- ^6^	[34]
SES5725	O80:H45	nd	nd	---	
SES6156		nd	nd	---	
Serotype collection	O80:H26 ^5^	nd	nd	---	This study
	O78:H4	nd	nd	d/d	

^1^ EH reference in [21]; SES reference is a laboratory new reference. ^2^ d: gene detected; nd: gene not detected. ^3^ AE-STEC O80:H2 EH2282 strain was erroneously referred to as *stx2a* instead of *stx1a* in Figure 1 of reference [21]. ^4^ AE-STEC O80:H2 EH2282 strain was the donor strain for the pS88 plasmid conjugation; AE-STEC O80:H2 EH3160 strain was the donor strain for STX2d phage transduction. ^5^ AE-STEC O80:H2 EH3320/SES3090 and serotype collection *E. coli* O80 strains were previously tested in *G. mellonella* [30]. ^6^ --- = no pS88 plasmid.

**Table 2 vetsci-10-00420-t002:** Target genes, primer sequences and amplified fragment lengths of the PCR applied on the DH10B pS88 plasmid transconjugant candidates.

PCR	Target Genes	Primers	Amplified Fragment Length	Reference
Serotype O80	*wzy_O80_*	Og80-F: 5′-TGGTGTTGATTCCACTAGCGT-3′ Og80-R: 5′-CGAGAGTACCTGGTTCCCAAA-3′	285 bp	[40]
Serotype H2	*fliC_H2_*	Hg2-F: 5′-TGATCCGACACTTCCTGATG-3′ Hg2-R: 5′-CCGTCATCACCAATCAACGC-3′	228 bp	[41]
Serotype O16	*wzx_O16_*	Og16-F: 5′-GGTTTCAATCTCACAGCAACTCAG-3′ Og16-R: 5′-GTTAGAGGGATAATAGCCAAGCGG-3′	302 bp	[40]
Colicin V_wt	*cvaC*_wt	CvaC-F: 5′-TATGAGAACTCTGACTCTAAAT-3′ CvaC-R: 5′-ATTTATAAACAAACATCACTAA-3′	314 bp	[42]
Colicin V_M ^1^	*cvaC*_M		1555 bp ^1^	
Avian hemolysin	*hlyF*	HlyF-F: 5′-GGCGATTTAGGCATTCCGATACTC-3′ HlyF-R: 5′-ACGGGGTCGCTAGTTAAGGAG-3′	599 bp	[19]

^1^ The *cvaC* gene is disrupted by the insertion of an ampicillin resistance cassette (see Section 2.3).

**Table 3 vetsci-10-00420-t003:** Virulence genes detected after genome analysis with Virulence Finder 2.0 of the *E. coli* O80:H2, O80:H6, O80:H26, O80:H45 and O78:H4 strains.

*E. coli* Serotypes and Strains		O80:H2 ^1,2^								O80:H6 ^3^	O80:H26 ^4^	O80:H45 ^3^		O78:H4 ^4^
Genes Detected after Genome Analysis		SES5320	SES5363	EH2282 ^5^	EH3307/ SES2959	EH3320/ SES3090	EH3160	EH3308/ SES2973	EH3322/ SES3122	SES6039	EH3161	SES5725	SES6156	O78C
Phage-located genes	*stx1a*	+ ^6^	+	+										
	*stx2d*				+	+	+							
LEE-located genes	*eaeξ*	+	+	+	+	+	+	+	+					
	*espA/B/F/P*	+	+	+	+	+	+	+	+					
	*tir*	+	+	+	+	+	+	+	+					
pS88 plasmid-located genes	*cia*	+	+	+	+	+	+	+	+					+
	*cvaA*	+	+	+	+	+	+	+	+					+
	*hlyF*	+	+	+	+	+	+	+	+					+
	*iroN*	+	+	+	+	+	+	+	+					+
	*iss*	+	+	+	+	+	+	+	+	+		+	+	+
	*ompT*	+	+	+	+	+	+	+	+			+	+	+
	*sitA*	+	+	+	+	+	+	+	+			+	+	+
	*etsC*		+	+		+	+		+					+
	*iucC*		+	+		+	+		+					+
Chromosome-located genes	*cma*	+		+	+			+						
	*iha*	+		+	+			+						

^1^ all but *stx1a* AE-STEC SES5320 and SES5363 strains were previously sequenced [21]. The Genbank accession numbers are SAMN35130764 (SES5320) and SAMN35130765 (SES5363). ^2^ EH numbers are listed in [21]; SES numbers are the new laboratory collection references. ^3^ all three *E. coli* O80:H6 and O80:H45 strains were previously sequenced [34]. ^4^ The Genbank accession numbers are SAMN35130763 (EH3161) and SAMN35130762 (O78C). ^5^ *stx1a* AE-STEC EH2282 strain was erroneously referred to as *stx2a* instead of *stx1a* in Figure 1 of reference [21]. ^6^ +: gene detected; empty box: gene not detected.

## Data Availability

The data were submitted to NCBI GenBank under accession numbers SAMN35130762; SAMN35130763; SAMN35130764; SAMN35130765.

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
