# Peer review of "Virulence of Shigatoxigenic and Enteropathogenic Escherichia coli O80:H2 in Galleria mellonella Larvae: Comparison of the Roles of the pS88 Plasmids and STX2d Phage"

_vetsci, 2023, doi:10.3390/vetsci10070420_

Round 1
Reviewer 1 Report
The manuscript: Virulence of Shigatoxigenic and Enteropathogenic Escherichia 2 coli O80:H2 in Galleria mellonella Larvae: Comparison of the 3 Roles of the pS88 Plasmids and STX2d Phage by Rie Ikeda and co-authors addresses essential aspects related to the pathogenicity mechanism of E. coli species. The authors analyse the possibilities of using Galleria mellonella larvae as a model for assessing the pathogenicity of the Shigatoxigenic and enteropathogenic Escherichia coli O80:H2, including the respective roles of two virulence properties: the pS88 plasmid-encoded invasiveness properties and the phage-encoded Shigatoxin 2d. In pursuit of the research goal, they use sequential analyses and methods of obtaining transconjugants and transductants.
My comments are as follows:
- Chapter materials and methods - the authors try to describe the strains and methods very precisely, referring to the results previously published, which in turn causes some chaos and is difficult to understand.
1. regarding 1.1. Bacterial strains – line 133 - E. coli K12 DH5α strain as positive and negative control – please specify in which tests/analyses
2. regarding 1.2. Genome sequencing - line155 - this first sentence needs correction. It is important to highlight those strains sequenced in this study and with previously published data at the end. Table 1 clearly shows that four strains were sequenced here.
3. Regarding 1.3. Construction of DH10B transconjugant and transductant
– line174-184 - characterisation of antibiotic susceptibility patterns of a collection of 33 strains has already been published and is not needed. Synthetic antibiotic resistance information for the AE-STEC EH2282 strain should be provided to justify its selection as a plasmid donor in conjugation.
- units of concentration (uppercase, lowercase, italics) need to be standardised - they should be adjusted according to the instructions for authors
- misspellings - lines 168, 222
4. regarding 2.4. In vivo assay: the Galleria mellonella model - line 224 - "all 17 E. coli strains" - what are these strains, and where did these strains come from? The analysis is supposed to cover 11 strains, right?
- Chapter of Results
1. regarding 3.3. Virulence of E. coli strains in G. mellonella larvae: Explain why strains 080:H26 and O78:H4 were used as control and which one was positive or negative control; refer to Table 3.
- Chapter Discussion - line 530 - The sentence needs correction, it's not clear, and it's an important conclusion
- Conclusions: line 633-634 - the sentence requires linguistic correction
General note - there are many “mental shortcuts” in the text. The work requires linguistic correction.
Author Response
REVIEWER #1
The manuscript: Virulence of Shigatoxigenic and Enteropathogenic Escherichia 2 coli O80:H2 in Galleria mellonella Larvae: Comparison of the 3 Roles of the pS88 Plasmids and STX2d Phage by Rie Ikeda and co-authors addresses essential aspects related to the pathogenicity mechanism of E. coli species. The authors analyse the possibilities of using Galleria mellonella larvae as a model for assessing the pathogenicity of the Shigatoxigenic and enteropathogenic Escherichia coli O80:H2, including the respective roles of two virulence properties: the pS88 plasmid-encoded invasiveness properties and the phage-encoded Shigatoxin 2d. In pursuit of the research goal, they use sequential analyses and methods of obtaining transconjugants and transductants. My comments are as follows:
- Chapter materials and methods - the authors try to describe the strains and methods very precisely, referring to the results previously published, which in turn causes some chaos and is difficult to understand.
- regarding 1.1. Bacterial strains – line 133 - E. coli K12 DH5α strain as positive and negative control – please specify in which tests/analyses
The section 2.1 was modified to make it clearer which strain was a positive or a negative control in larvae of Galleria mellonella and the reason for.
- regarding 1.2. Genome sequencing - line155 - this first sentence needs correction. It is important to highlight those strains sequenced in this study and with previously published data at the end. Table 1 clearly shows that four strains were sequenced here.
The sections 2.1 and 2.2 were modified to meet the reviewer’s comment.
- Regarding 1.3. Construction of DH10B transconjugant and transductant
– line174-184 - characterisation of antibiotic susceptibility patterns of a collection of 33 strains has already been published and is not needed. Synthetic antibiotic resistance information for the AE-STEC EH2282 strain should be provided to justify its selection as a plasmid donor in conjugation.
Actually, the resistance profiles of these 33 strains have not been previously published. This is why we gave all details about the procedure. On the other hand, we did not find necessary to show the detailed results of all strains in this manuscript.
- units of concentration (uppercase, lowercase, italics) need to be standardised - they should be adjusted according to the instructions for authors
Done
- misspellings - lines 168, 222
Done
- regarding 2.4. In vivo assay: the Galleria mellonella model - line 224 - "all 17 E. coli strains" - what are these strains, and where did these strains come from? The analysis is supposed to cover 11 strains, right?
Actually, a total of 17 strainsn including 11 wild-type strains were tested in larvae of Galleria mellonella, a number that includes all controls and the transconjugant and transductant. The complete list was added between “()”.
- Chapter of Results
- regarding 3.3. Virulence of E. coli strains in G. mellonella larvae: Explain why strains 080:H26 and O78:H4 were used as control and which one was positive or negative control; refer to Table 3.
The reason of choosing the E. coli O80:H26 is now more precisely explained in section 2.1 while the reason of choosing the E. coli O78:H4 is now introduced in section 2.1 and explained in section 3.1. The initial reason is the published literature showing that the serotype O78 is highly pathogenic, especially in poultry. Our choice was confirmed by the results of preliminary testing in larvae of Galleria melonella. So we had: one K12 negative control, one non-O80 positive control, one O80:non-H2 positive control.
- Chapter Discussion
- line 530 - The sentence needs correction, it's not clear, and it's an important conclusion
This sentence was modified and completed.
- Conclusions:
- line 633-634 - the sentence requires linguistic correction
Correction made: partially èpartly. Sorry about this linguistic mistake.
- Comments on the Quality of English Language. General note - there are many “mental shortcuts” in the text. The work requires linguistic correction.
Since the delay was very short to submit a revised manuscript (5 days) and since the second reviewer did not ask any revision of the English language, no other linguistic correction was made.
Reviewer 2 Report
This manuscript uses Galleria mellonella larvae as a model system to compare the relative virulence of different E. coli strains and the contributions of specific factors (Shiga toxins, p88 plasmid) to virulence. The experimental design and data interpretation are straightforward, with appropriate consideration of the limitations of the study. I suggest one minor modification to improve the clarity of the work:
Four genes (cma, iha, etsC, and iucC) are mentioned repeatedly in the text, but only the gene product for iucC is defined – and only at the end of the paper, in the Discussion. To help the reader understand the relevance of these genes, the Introduction should explain how all four gene products contribute to virulence.
Typos:
Italicize E. coli on lines 49 and 623
uG should be ug on lines 196 and 204
“carrying the” seems to be out of place at the end of line 555.
Author Response
REVIEWER #2
This manuscript uses Galleria mellonella larvae as a model system to compare the relative virulence of different E. coli strains and the contributions of specific factors (Shiga toxins, p88 plasmid) to virulence. The experimental design and data interpretation are straightforward, with appropriate consideration of the limitations of the study.
- i) I suggest one minor modification to improve the clarity of the work: “Four genes (cma, iha, etsC, and iucC) are mentioned repeatedly in the text, but only the gene product for iucC is defined – and only at the end of the paper, in the Discussion. To help the reader understand the relevance of these genes, the Introduction should explain how all four gene products contribute to virulence.”
Thank you for this suggestion. However, we think that the discussion about the (putative) role of these four genes in E. coli pathogenicity of O80:H2 strains in larvae of Galleria mellonella is more appropriate in the DISCUSSION than in the INTRODUCTION. Following the reviewer’s suggestion meant that 4 references were added to the list.
- ii) Italicize colion lines 49 and 623
Done
iii) uG should be ug on lines 196 and 204
Done
- iv) “carrying the” seems to be out of place at the end of line 555.
Done